# Active machine learning-driven experimentation to determine compound effects on protein patterns

Armaghan W Naik[1,2], Joshua D Kangas[1,2], Devin P Sullivan[1,2], Robert F Murphy[1,2,3,4,5,6,7]*

[1]Computational Biology Department, Carnegie Mellon University, Pittsburgh, United States; [2]Center for Bioimage Informatics, Carnegie Mellon University, Pittsburgh, United States; [3]Department of Biological Sciences, Carnegie Mellon University, Pittsburgh, United States; [4]Department of Biomedical Engineering, Carnegie Mellon University, Pittsburgh, United States; [5]Machine Learning Department, Carnegie Mellon University, Pittsburgh, United States; [6]Freiburg Institute for Advanced Studies, Albert Ludwig University of Freiburg, Freiburg, Germany; [7]Faculty of Biology, Albert Ludwig University of Freiburg, Freiburg, Germany

**Abstract** High throughput screening determines the effects of many conditions on a given biological target. Currently, to estimate the effects of those conditions on other targets requires either strong modeling assumptions (e.g. similarities among targets) or separate screens. Ideally, data-driven experimentation could be used to learn accurate models for many conditions and targets without doing all possible experiments. We have previously described an active machine learning algorithm that can iteratively choose small sets of experiments to learn models of multiple effects. We now show that, with no prior knowledge and with liquid handling robotics and automated microscopy under its control, this learner accurately learned the effects of 48 chemical compounds on the subcellular localization of 48 proteins while performing only 29% of all possible experiments. The results represent the first practical demonstration of the utility of active learning-driven biological experimentation in which the set of possible phenotypes is unknown in advance.

*For correspondence: murphy@cmu.edu

**Competing interests:** The authors declare that no competing interests exist.

## Introduction

Classical screening methods determine the phenotype of a biological component under many conditions (such as the presence of different mutations or the addition of small molecules or inhibitory RNAs). What phenotypes would be elicited by these conditions for other biological components is unknown unless either additional screens are performed, or it is already known how effects on one component generalize to others. The risk of not generalizing can be great; drug candidate screens identify compounds that perturb a particular target in a desired way, but their possible off-target or side-effects are not measured during the screening process and are often only discovered late in drug development (*Lounkine et al., 2012*; *Macarron et al., 2011*; *Trist, 2011*). In principle, better drug candidates could be discovered by performing experiments for every combination of potential target and condition. However, exhaustive experimentation is infeasible for essentially all biological systems (*Murphy, 2011*). These indicate the need for a practical method for both iteratively choosing a subset of the total experiment space to observe, and for generalizing the observed results to a potentially much larger set. This type of approach is referred to as *active learning* in the machine learning literature.

**eLife digest** Biomedical scientists have invested significant effort into making it easy to perform lots of experiments quickly and cheaply. These "high throughput" methods are the workhorses of modern "systems biology" efforts. However, we simply cannot perform an experiment for every possible combination of different cell type, genetic mutation and other conditions. In practice this has led researchers to either exhaustively test a few conditions or targets, or to try to pick the experiments that best allow a particular problem to be explored. But which experiments should we pick? The ones we think we can predict the outcome of accurately, the ones for which we are uncertain what the results will be, or a combination of the two?

Humans are not particularly well suited for this task because it requires reasoning about many possible outcomes at the same time. However, computers are much better at handling statistics for many experiments, and machine learning algorithms allow computers to "learn" how to make predictions and decisions based on the data they've previously processed.

Previous computer simulations showed that a machine learning approach termed "active learning" could do a good job of picking a series of experiments to perform in order to efficiently learn a model that predicts the results of experiments that were not done. Now, Naik et al. have performed cell biology experiments in which experiments were chosen by an active learning algorithm and then performed using liquid handling robots and an automated microscope. The key idea behind the approach is that you learn more from an experiment you can't predict (or that you predicted incorrectly) than from just confirming your confident predictions.

The results of the robot-driven experiments showed that the active learning approach outperforms strategies a human might use, even when the potential outcomes of individual experiments are not known beforehand. The next challenge is to apply these methods to reduce the cost of achieving the goals of large projects, such as The Cancer Genome Atlas.

We have previously described an active learning method applicable to large sets of targets and conditions (*Naik et al., 2013*). Using simulations, we showed how a predictive model could be learned by incrementally selecting sparse subsets of experiments to perform based on the results from previous sets. The results showed that accurate predictions (of whether a given drug would affect a given target) were learned significantly more rapidly when this active learner was used to guide sequential experiment selection than when experiments were selected at random. The critical assumption of that work was that once an experiment was performed it could be unambiguously assigned to one of a set of known phenotypes. While this may be a reasonable assumption for some cases (e.g. on/off expression phenotypes), for most drug screening systems it is not only difficult to define distinct phenotypes but the general problem of inferring which phenotypes are possible by clustering observations is considered to not have a solution (*Kleinberg, 2002*; *Vapnik, 1998*). Thus in order to use active learning for complex, real-world scientific applications, we must demonstrate its feasibility under conditions where the number and types of phenotypes must be estimated as data are acquired during the learning process.

In this paper, we consider the problem of using active learning to determine how multiple proteins change their subcellular location patterns in response to multiple chemical compounds. To demonstrate the feasibility of our approach to this problem, we performed a pilot study using a small and spatially diverse set of proteins to capture the effects of a modest number of drugs on different subcellular structures (since we lacked the resources to consider all proteins and a large drug library). Note that our goal is to identify whether a given drug perturbs the pattern of a given protein, and symmetrically, which drugs perturb which proteins in a similar manner. In doing so, we do not seek to describe each protein or type of perturbation in terms of a previously described organelle or structure, since previous work has illustrated that some protein patterns are not typical of any single organelle (*Chen and Murphy, 2005*; *Chou et al., 2011*), and some perturbations may not have been previously observed (and therefore not yet named). Similar to the approach taken in screening drug libraries, we considered a small and chemically diverse set of perturbagens in hopes of identifying salient patterns of effects (*Inglese et al., 2007*; *Macarron et al., 2011*). While there is

a large literature on chemical library design (*Gordon et al., 1994*; *Welsch et al., 2010*), some of which attempts to make use of observed data or design of experiments (*Tye, 2004*), we are unaware of methods which have been applied to studying how the behavior of large numbers of *targets* beyond single classes (e.g. kinases, GPCRs, etc.) are affected.

Our approach is similar to other high-content campaigns (*Abraham et al., 2004*; *Zanella et al., 2010*) in that we made extensive use of liquid handling robotics for both drug manipulation and cell culture. The crucial distinctions and novelty of this work are that multiple targets and perturbagens were considered at the same time and that the experiment loop (deciding what experiments to perform next) was entirely guided by a machine learning algorithm without human intervention. While active learning and similar ideas have been applied to biological data as post-hoc or retrospective analyses (*Danziger et al., 2009*; *Liu, 2004*; *Mohamed et al., 2010*; *Romero et al., 2013*) and while robotically-executed experiments have been carried out (*King et al., 2009*), to our knowledge this is the first series of active learning-driven *prospective* biological experiments where the possible answers (e.g., what phenotypes might be observed) were not known *a priori* with the only constraint being the type of experiment that could be performed.

## Results

### Experiment space construction and active learning

We have previously constructed an atlas of unperturbed protein subcellular location patterns by extensive CD-tagging in NIH-3T3 cells (*Coelho, 2013*; *Garcia Osuna et al., 2007*) which produced clones endogenously expressing different EGFP tagged proteins. From fluorescent microscopy images of these cells we chose 48 different clones (*Supplementary file 1*) collectively representing a broad range of location patterns (*Figure 1*). We chose an additional six clones, distinct from the above, for independent testing of how well a model learned from the 48 would generalize to unobserved proteins. We also formed a library of 48 different treatment conditions ('drugs"') (*Table 1*): 47 chemical compounds suspected to affect some aspect of subcellular trafficking, structure or localization, together with a vehicle-only control (no drug). The clones and drugs were each assigned numbers by which the active learner could refer to them; both the clones and drugs were "duplicated" by assigning two numbers to each, and this duplication was hidden from the learner. The learner was thus presented with a 96 x 96 space of possible experiments in which an experiment consisted of acquiring images for a given clone in the presence of a given drug. As described in Materials and methods, a new experiment was done when the learner requested it even if an experiment had been done previously for a combination of either of the duplicates of that drug and target – images were not shared across the duplicates and therefore the learner could not easily uncover the duplications by seeing which images were the same. The rationale for the duplication was to provide a basis for evaluating the choices of experiments by the learner after active learning was completed.

The first *round* of experiments began by collecting images of all clones for one of the vehicle-only conditions (96 experiments). For analysis and model building, images were represented by numerical features that captured the subcellular localization of the EGFP labeled protein. Data from each experiment were subjected to a quality control procedure established at the outset from the initial data (see Materials and methods).

At the end of each round (including the first), all experiments up to and including that round (that passed quality control) were used to identify *phenotypes*. We use the term 'phenotype' to refer to a statistically distinguishable localization pattern that may or may not correspond to a previously described or characterized drug effect; see Discussion. Each experiment was represented by a set of feature vectors corresponding to its images. We clustered these sets such that clusters of experiments with similar feature vectors defined phenotypes (see Materials and methods). The number of phenotypes was determined anew from the data every round; which patterns are statistically different may change from round to round as new images might contain entirely new patterns, might be additional examples of a pattern that was previously not considered statistically significant, or might show a pattern that is intermediate between two previous clusters causing them to become joined. The phenotype assignments were then given to the learner to form a model to make predictions about unmeasured experiments (see Materials and methods). The basis of this predictive model was

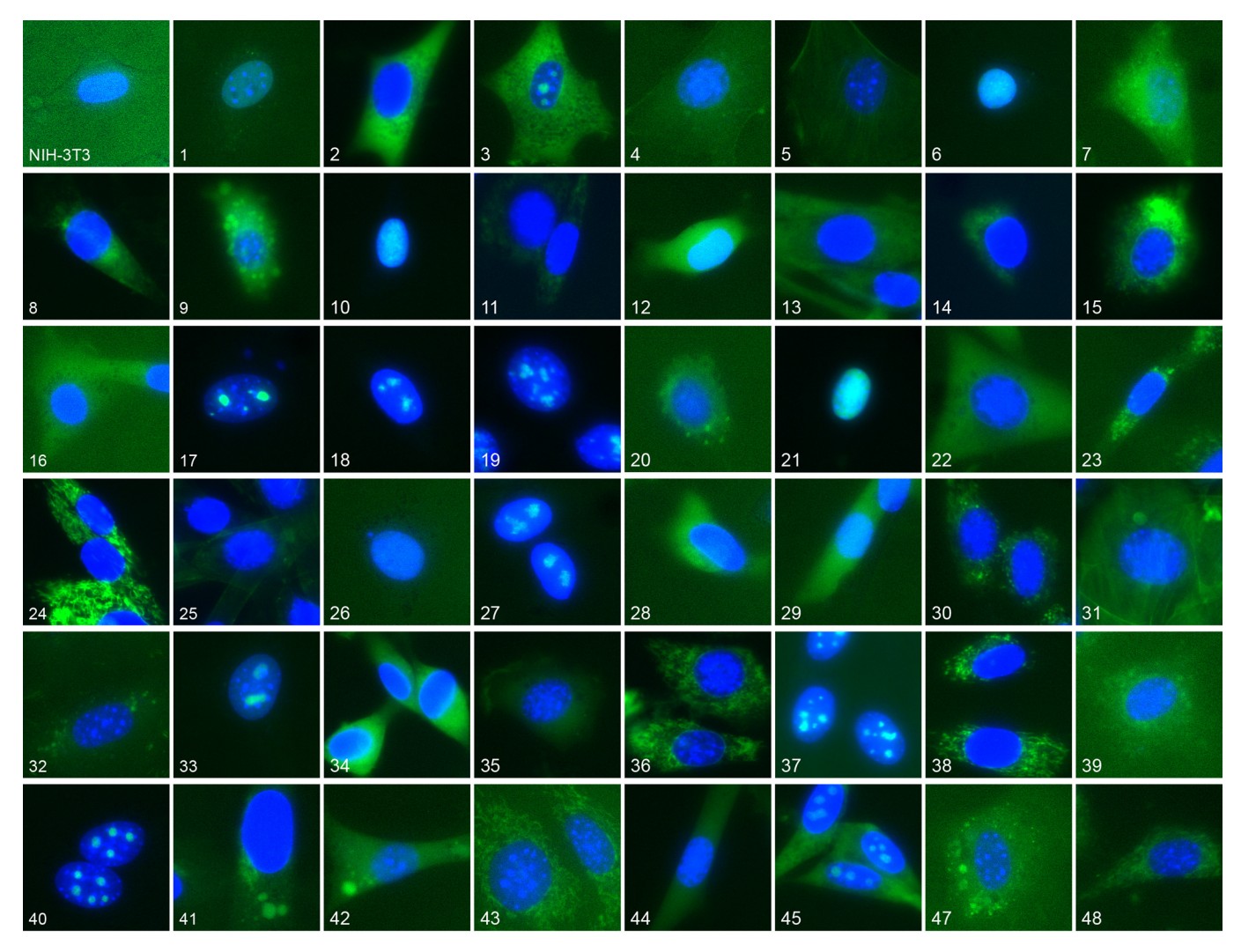

**Figure 1.** Representative location patterns of the CD-tagged clones. Images of EGFP (green) and Hoechst 33,342 (blue) fluorescence were acquired at 40x with an automated widefield microscope (see Materials and methods). Each panel is independently contrast stretched. The identities of the tagged gene for each clone are listed in *Supplementary file 1*. Clone order is random with respect to location pattern. The untagged NIH 3T3 (upper left) was assigned as clone 46.

to group together those drugs that had been observed to elicit the same phenotype for at least one clone and had not been observed to elicit different phenotypes for any clone (and similarly by grouping clones). This grouping reduced the complexity of the problem: each group of drugs or clones was assumed to behave the same; of course, later experiments might reveal that some of these groupings were incorrect. The predictions of the model were therefore that drugs in the same group would show the same effect on all clones in future experiments (and that all clones in the same group would be affected similarly). Using this model of correlations of phenotypes among drugs and clones, the active learner chose a new round of 96 new experiments to be performed. The choice of experiments in a round simultaneously prioritized experiments that would test the greatest number of the groupings (i.e., experiments predicted by the largest groups of drugs or clones) while minimizing the number of experiments that could be predicted from each other (i.e., experiments predicted by the same group).

The learner repeated this process for a total of 30 rounds (this number was chosen due to budgetary constraints). Since not all experiments passed quality control, 30 rounds accounted for 2697

**Table 1.** Compounds used.

| Compound # | Compound | Stock concentration (mM) (in 100% DMSO) |
|---|---|---|
| 1 | Apicidin | 2.00 |
| 2 | Cytochalasin D | 2.45 |
| 3 | Latrunculin B | 1.25 |
| 4 | Cycloheximide | 1.75 |
| 5 | α-amanitin | 0.25 |
| 6 | Camptothecin | 2.85 |
| 7 | Chloramphenicol | 12.4 |
| 8 | Nocodazole | 8.3 |
| 9 | Diethylstilbestrol | 1.85 |
| 10 | Dinitrophenol | 35.3 |
| 11 | Griseofulvin | 4.95 |
| 12 | Amiloride hcl | 1.88 |
| 13 | Alsterpaullone | 0.85 |
| 14 | Dimenhydrinate | 13.85 |
| 15 | Colchicine | 3.75 |
| 16 | Econazole | 3.35 |
| 17 | Chloroquine | 1.45 |
| 18 | Bulsulfan | 6.1 |
| 19 | Actinomycin D | 1.2 |
| 20 | Radicicol | 1.35 |
| 21 | Calmidazolium | 1.45 |
| 22 | Etoposide | 0.85 |
| 23 | z-Leu(3)-Al | 1.05 |
| 24 | Exo2 | 1.4 |
| 25 | Exo2 | 0.7 |
| 26 | Exo2 | 0.35 |
| 27 | Brefeldin A | 0.9 |
| 28 | Brefeldin A | 0.45 |
| 29 | Brefeldin A | 0.23 |
| 30 | Cytochalasin D | 1.23 |
| 31 | Cytochalasin D | 0.61 |
| 32 | Latrunculin B | 0.63 |
| 33 | Latrunculin B | 0.31 |
| 34 | Staurosporine | 0.55 |
| 35 | Leptomycin B | 0.025 |
| 36 | Trichostatin A | 0.025 |
| 37 | Paclitaxel | 0.645 |
| 38 | Ganciclovir | 3.15 |
| 39 | Monensin | 1.1 |
| 40 | 5-azacytadine | 2.85 |
| 41 | Na butyrate | 2.95 |
| 42 | Hydroxyurea | 22.35 |
| 43 | Clonidine hcl | 4.3 |

*Table 1 continued on next page*

Table 1 continued

| Compound # | Compound | Stock concentration (mM) (in 100% DMSO) |
|---|---|---|
| 44 | Alginate lysate | N/A * |
| 45 | Leptomycin B | 1.0125 |
| 46 | Trichostatin A | 0.125 |
| 47 | mdivi-I | 3.55 |
| 48 | Vehicle (No Drug) | N/A |

1.3 µL of a given stock were added to 1000 µL, so the final concentrations used are 1/1000 of the concentration listed.

* 2.2 mg of the lysate (from *Flavobacterium multivorum*, Sigma-Aldrich) was dissolved in 2.0 mL DMSO to form the stock solution; units indeterminate.

experiments (~29%) of the total 96 x 96 experiment space, which covered 1670 experiments (~72%) in the underlying (unduplicated) 48 x 48 experiment space. During the process of data collection, there was no human intervention as to which experiments to perform, nor did we attempt any assessment of the performance of the learner. To complete the dataset, the 634 combinations of drug and clone (ignoring duplication) that had not been chosen by the learner were collected after the 30 rounds. We also collected images for all drugs for six clones outside the set of 48 that the learner knew about.

## Accuracy of learning

After data collection had been completed, we first asked whether the active learner accurately predicted phenotypes for unobserved experiments. We assessed the accuracy of the predictions of a model from a given round using data it had not yet seen: data from the 'completion' experiments as well as from experiments that were performed in subsequent rounds. Each prediction for an experiment consisted of a phenotype from the set of phenotypes observed so far, and was considered correct if a plurality of images from that experiment were closest (in the feature space) to an image from an observed experiment that had been assigned to the same phenotype. That is, for each of the images from an as yet unobserved experiment, the image was found that was closest to it out of all that the learner had already observed. The phenotype that this closest image had been assigned by the learner was given to the image from the unobserved experiment. The number of unobserved images assigned to each phenotype was then counted, and if more images were assigned to the predicted phenotype than any other individual phenotype, the prediction was considered correct (note that the phenotypes themselves may change from round to round). This definition parallels the nearest-neighbor methods used for clustering (see Materials and methods). Using this definition, the accuracy of the model learned after each round was retrospectively calculated (*Figure 2*, black line).

We then asked whether the last actively learned model could make accurate predictions for the six clones the learner had not seen. Given just the images for the unperturbed (no-drug) condition for the six clones, we generated predictions for the remaining 47x6 experiments and assessed them using the same nearest neighbor plurality approach described above. As a baseline accuracy for these predictions, we can consider what predictions we could make without having learned a model. In this case, we have no way of knowing what phenotypes are possible (other than the unperturbed phenotypes), and must predict that no matter what drug is added the phenotype would remain unperturbed. For the six unseen clones, this would lead to an accuracy of 85%, i.e., 15% of the experiments showed a phenotype different from the unperturbed phenotype for that clone. The model from active learning did much better than this, giving an accuracy of 98% (only 5–6 out of 282 experiments were incorrectly predicted). As can be expected, the learner's predictions were also significantly better than expected for random guessing if we are given the set of possible phenotypes ($P<0.05$, Multinomial test). This high accuracy at generalizing to new clones suggests that the final model had captured quite well the effects of the drugs on the localization of any target (at least for targets somewhat similar to the original set).

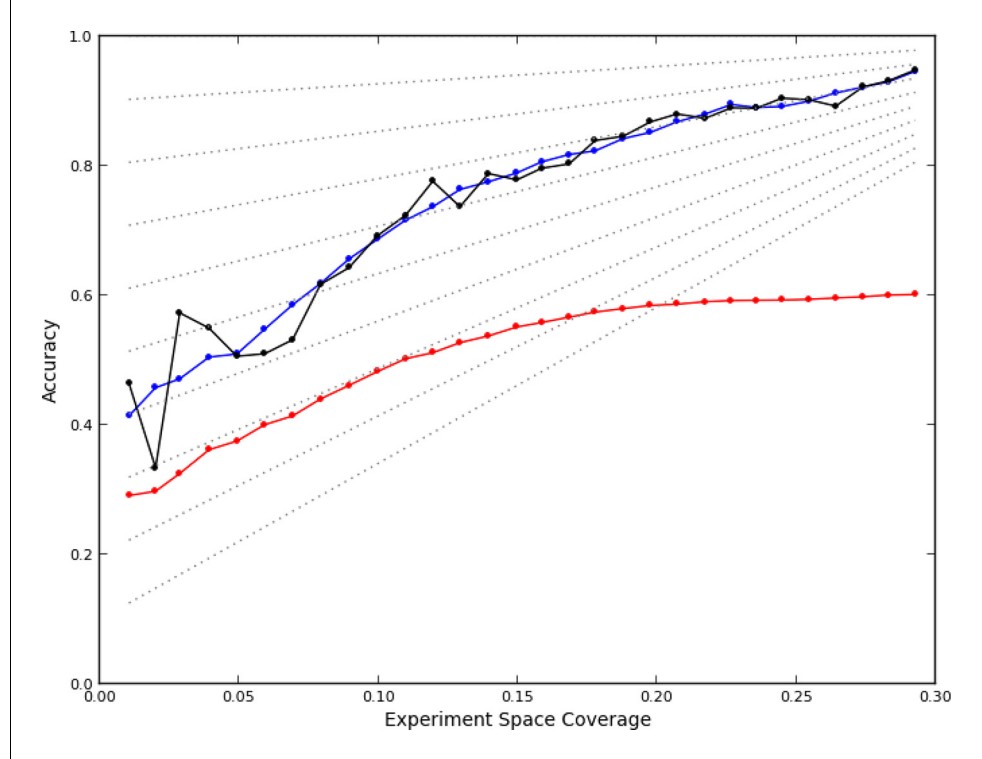

**Figure 2.** Accuracy of active learning. The performance of the active learner (black line) generally increased superlinearly as more data were acquired. Hypothetical models (dotted gray lines) with fixed generalization accuracy have constant slopes and are displayed for reference (10% to 90% rates as isoclines). The initial model poorly generalized (~45%) while the final model learned at round 30 (29% experiment space coverage) had ~75% generalization accuracy and 92% overall accuracy. A regression model based on unique experiment coverage (blue line, see main text for details) qualitatively explains the observed learner performance. Using this coverage model, an estimate of expected accuracy for random learning was constructed (red line, see main text for details); the final accuracy difference between the active learner and random learning is ~40%.

## Efficiency of learning

Due to the duplications, for every unique combination of clone and drug, the active learner could have selected to perform up to four logically equivalent (i.e. duplicated) experiments; we refer to these sets of four as *quads*. The learner should be able to learn these (hidden) equivalences as its learning proceeds, and eventually avoid performing multiple experiments in the same quad. To illustrate this process, *Video 1* shows, for each round, how many experiments were done for each quad and the accuracy of the predictions made at that round for experiments that had not yet been performed (assessed in the same manner as for *Figure 2*). To calibrate these results, in an optimistic setting an accurate model could be learned by performing only ~26% of the 96 x 96 experiments (see Materials and methods). Presumably an *efficient* learner should learn regularities quickly and then avoid excessive sampling of each of the (48 x 48 = 2304) quads. In other words, once the learner had realized that clone A was similar to clone B, it could do experiments for only one of those clones and then predict that the same result would have obtained for the other. The same holds for recognizing that two drugs are similar. Thus, the extent to which the learner chose to do experiments for only one clone and one drug from the same quad reflects its ability to recognize similarities and thus do experiments efficiently.

Intuitively, when the learner chose an unmeasured quad to examine, if the drug elicited a statistically dissimilar localization pattern than was predicted, then the next model would likely be improved. Potentially, a different grouping of drugs and clones would have to be formed to explain the observed data, or new phenotypes would be estimated (since the data would have different statistically distinguishable parts), or a combination of these. We therefore asked how well

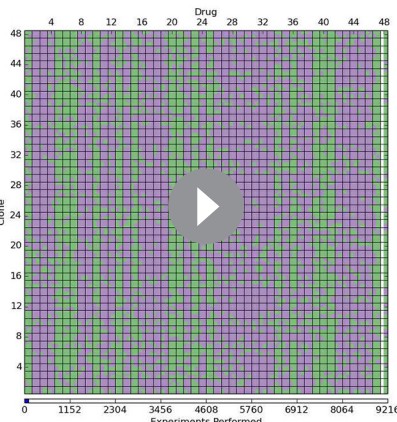

**Video 1.** Experiment selection and accuracy of predictions during the active learning process. Each drug and clone were duplicated in a manner hidden to the active learner (96 x 96 experiments) and are grouped for display purposes together (as 48 x 48). These four biological replicates (which we call a "quad") are outlined in black, and each is shown separately as a sub-box (not outlined) in its quad. The first frame shows the starting point: unperturbed experiments were measured for all clones (white boxes in a single subcolumn for drug 48) and the model predicted that all drugs lead to the same phenotypes. Each model's classification accuracy on unseen data is displayed for each 96 x 96 experiment (sub-boxes) in green when correct, and purple when not. For example, the first frame shows that the phenotypes for most drug treatments differed from their corresponding unperturbed condition, and so the overall accuracy was low (more purple than green). By design, the active learner chose the second batch of experiments to evenly sample each drug and clone (sparse white boxes). Those data led to a model with lower accuracy because emphasis was placed on ultimately spurious correlations in phenotypes. In general, the active learner always chose to perform experiments to test presumed correlations in phenotypes, and so there was a substantial increase in accuracy from round 2 to round 3. As additional rounds were performed, the accuracy gradually increased and most quads were only measured once. By round 20, many of the experiments had been correctly predicted and the learner focused on learning the remaining ones. By the last round, most predictions were correct, but the predictions for a few drugs remained largely incorrect.

generalization accuracy could be predicted from the degree to which quads were sampled. It is important to note the distinction between generalization accuracy and overall model accuracy: the former is the accuracy of predictions for unobserved experiments while the latter is the combination of the generalization accuracy and the percent of experiments done so far (since we assumed that the results of experiments that had been done were correct).

We summarized each round's experiments by constructing a five-bin histogram of how often experiments were done for the same quad: the first bin was the fraction of quads (out of 48 x 48 total) that were not sampled, the next bin was the fraction of quads that were sampled only once, and so on. For example, in round 11 there were 1529 quads never observed, 575 observed once, 164 observed twice, 36 observed three times, and none observed four times. This was encoded as the vector [0.66, 0.25, 0.07, 0.02, 0.0]. We fit the resulting 30 vectors, one for each round of learning, to the generalization accuracy at that round by linear regression (see Materials and methods). As shown in *Figure 2*, experiment coverage is a good predictor of observed model performance (blue line).

We can further understand the performance of the learner by examining the coefficients of regression, which are in units of expected accuracy (out of 1.0 or 100%) per coverage frequency. These were: 0.42 (quads not covered), 1.0 (covered once), -0.57 (2x), 6.4 (3x), and -21 (4x). For the round 11 data above, this fit predicts 60% generalization accuracy, which is close to the measured 61% (72% total accuracy).

This fit, as to be expected, shows a large opportunity cost for performing four experiments from the same quad that should have been recognized as being equivalent, and a slight penalty for performing two experiments. Interestingly, there was a benefit of performing three experiments in the same quad, although this happened rarely (7% of the time). We attribute this to the fact that some quads showed greater variation within their experiments than others (data not shown), leading the learner to do more experiments within a quad in order to improve its accuracy. The number of phenotypes generally increased as data were collected (*Figure 3*); part of this was due to learner assigning new phenotypes to account for these hypervariable experiments. Interestingly, the expected reward for not performing any experiments in a quad is positive; this is consistent with there being additional similarities beyond those resulting from the duplications (e.g., some of the clones or compounds are similar enough that some unmeasured quads can be accurately predicted from others).

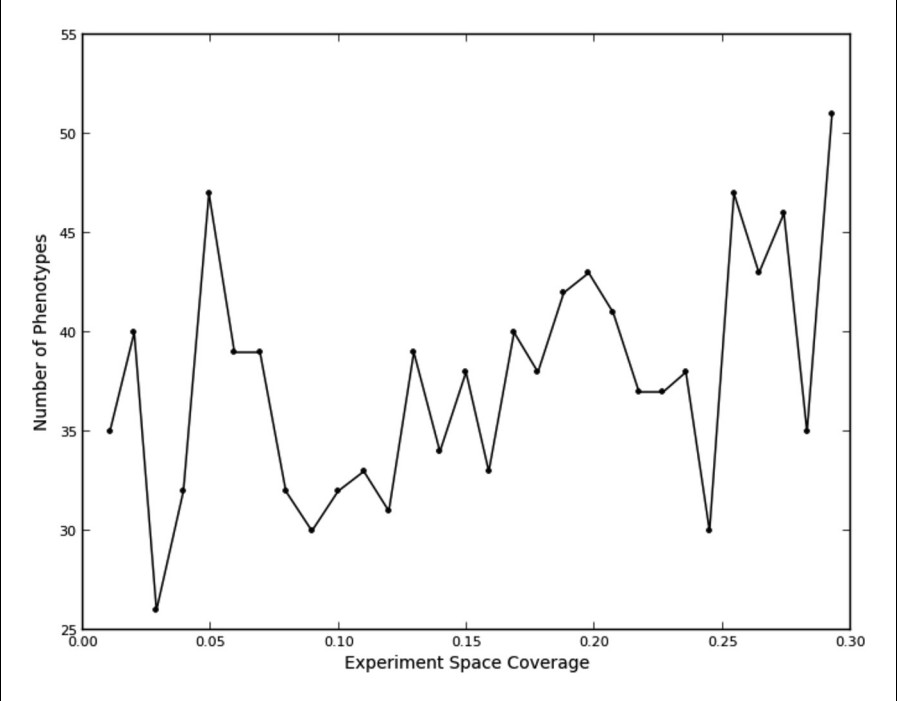

**Figure 3.** Number of phenotypes identified at each round in the learning process.

## Estimated accuracy from random learning

Active learning methods are often characterized in computer simulations relative to a learner that chooses experiments uniformly at random. We can construct an estimate of the expected performance of a random learner if it is assumed that, as with the actively learned data, generalization accuracy can also be predicted by the extent of quad coverage. This assumption avoids the complication (and computational expense) of having to simulate what data would have been acquired in the 96 x 96 space from the data that was actually acquired (i.e., generating new images or feature vectors).

We therefore simulated, at each round, how many experiments would have been chosen at random from each quad (i.e., how many balls randomly thrown into a 96 x 96 grid would have ended up in each quad of four bins). Applying the regression fit above to these randomly sampled 96 x 96 experiments (see Materials and mMethods) produces a lower rate of learning (*Figure 2*, red line); the final accuracy predicted for random sampling was 40% lower than that achieved by the active learner. While we cannot know that this estimate is correct (given the complicated process of phenotype estimation at each round), it is clear that the active learner chose highly nonrandom subsets of experiments to perform (*P*<0.05, analytical distribution, see Materials and methods) and that the resulting accuracy was much higher than expected from random choice.

## Identifying perturbations

Taken together, the previous analyses show that the active learner in able to accurately learn complex localization phenotypes, some of which may be quite subtle (see *Figure 4* for an overview). In other discovery contexts, such as drug screening, efficient identification of *acute* phenotypic changes is the goal. To accomplish this while incorporating the efficiency of active learning, we can imagine a process by which an active learner is first used to learn all phenotypes until some stopping criterion is met, and then the acute phenotypes are identified as a secondary task from the actively collected data. For this two-step process, we can apply more stringent quality control after data collection has ceased, in order to remove from consideration low quality images not caught by the initial automated quality control. We performed second step filtering as described in Materials and methods.

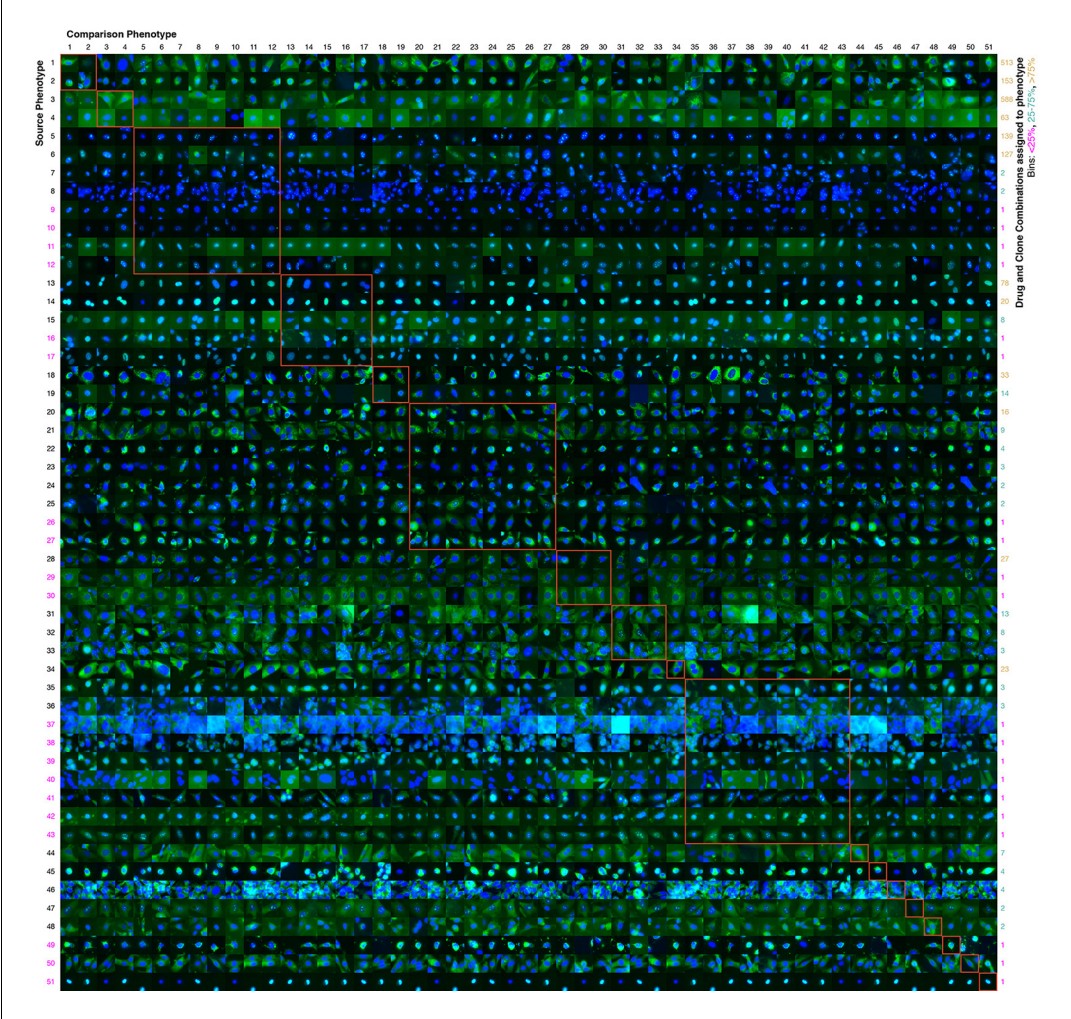

**Figure 4.** Contrasts between phenotypes identified by the active learner. The last actively learned model identified 51 'phenotypes,' each phenotype of which is defined by a set of imaged fields. To independently assess the extent to which these phenotypes were different, a logistic regression classifier was trained to distinguish the actively learned phenotypes and evaluated by cross-validation; the classifier was able to distinguish all 51 phenotypes in fields not used for training with 75% accuracy. To give a sense of the spread of each phenotype, a randomly chosen cell from a field in the source phenotype (row) that had the median classification accuracy against another phenotype (column) is shown; this field that is chosen can be considered representative of the source phenotype when considered relative to the other phenotype. In this way, visually across a row one sees examples from each phenotype reflective of differences between it and other phenotypes. Phenotypes have been reindexed (*Supplementary file 2* shows both indices for each drug-clone combination) and placed into groups to facilitate comparisons between visually similar phenotypes; within-group comparisons are outlined by orange squares (the human assigned labels corresponding to each group are shown in *Supplementary file 3*). Each phenotype was assigned to one or more drug-clone combinations; groups are ordered from most (top) to least (bottom) frequently assigned to experiments, and likewise within groups, phenotypes are ordered by frequency (right column, color coded by percentile bins: magenta for 1 experiment (25th percentile), cyan for 2–14 experiments (25–75th percentiles), and gold for the remainder). 20 phenotypes (39%) are assigned to a single combination of drug and clone; these account for just 1% of the combinations assessed by the learner. These rarely exhibit acute localization, and in only one case (phenotype 37) is this likely due to an experimental artifact (overly confluent fields). For example, in the third group from the top (mostly nucleolar localization), phenotype 9 appears to have condensed nucleolar localization relative to more popular phenotypes 5–8, and phenotype 10 appears to reflect smaller nuclei. Phenotype 11 contains some out-of-focus examples, but otherwise has greater cytosolic localization than the other nucleolar phenotypes. Phenotypes 35–43 appear to be enriched in cytotoxic responses, and include two phenotypes with confluent fields (36 and 37), however not all fields in those phenotypes are confluent. Some phenotypes are complex, such as phenotypes 20–27, which show a range of nominal secretory localization and cell body collapse or block in secretory localization. In general, cells sampled within phenotypes (across rows) are more visually similar to each other than between phenotypes, and phenotype differences are generally due to bona fide (albeit often subtle) localization differences rather than artifacts. The figure is best viewed on a computer to allow zooming; a full resolution version of the figure (400 MB) is available at http://murphylab.web.cmu.edu/software/2016_eLife_Active_Learning_Of_Perturbations/Figure4Full.pdf.

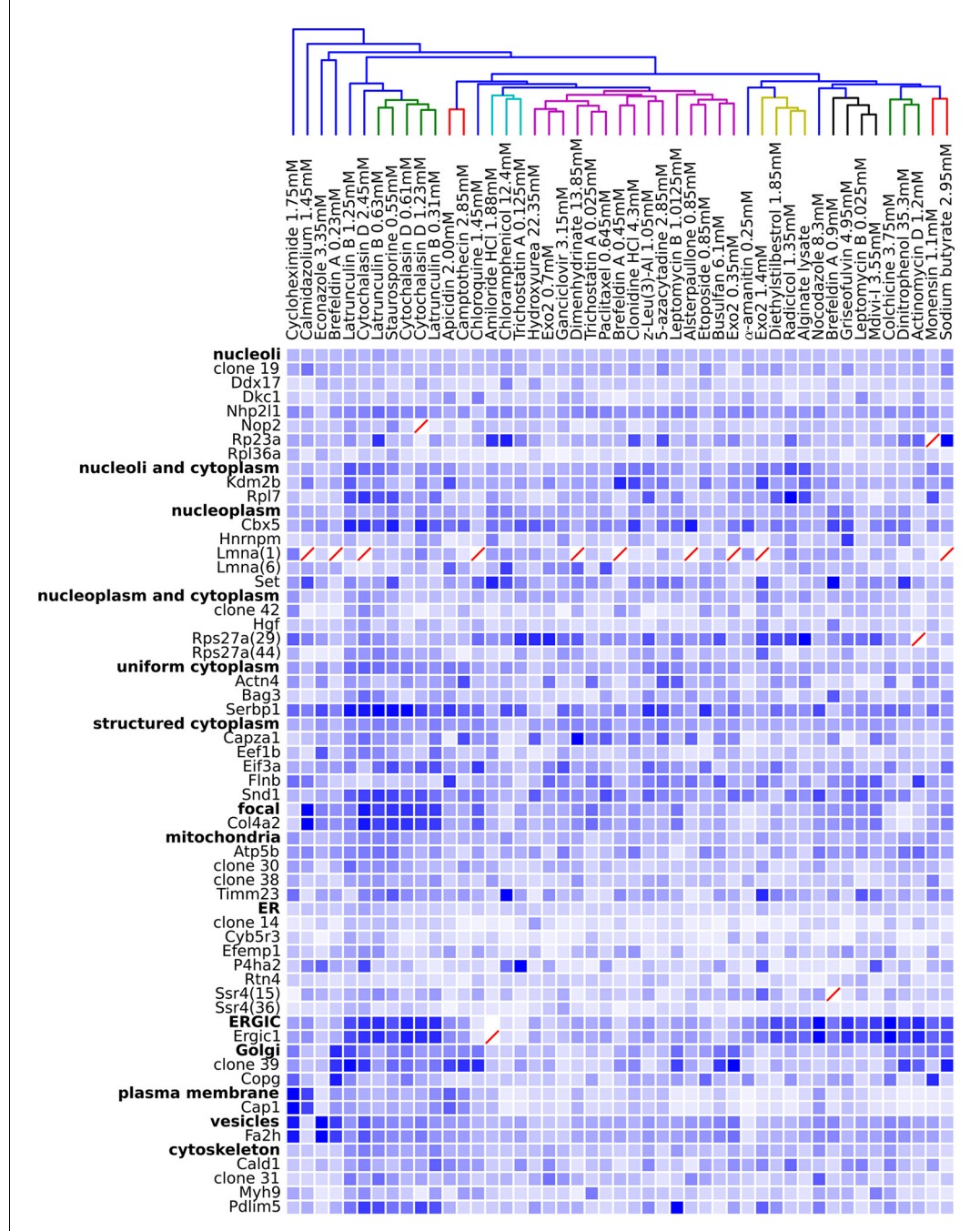

**Figure 5.** Degree of perturbation for all experimental combinations. The amount of perturbation for each combination of drug and clone is shown, with deeper shades of blue indicate larger degrees of perturbation (larger distances from the mean feature values for a clone and drug combination to the mean feature values of the vehicle-only control for that clone; feature values are contained in *Supplementary file 4*). Images were subjected to additional quality control for this analysis; diagonal red lines mark experiments failing this stricter quality control. Clones are grouped by the labels assigned to the unperturbed (control) subcellular localization patterns; the mean perturbation of all proteins with a given label is also displayed for each drug. Drugs were clustered with average linkage using the mean perturbation data. Different tagged variants of the same protein (labeled with clone identifier in parenthesis) sometimes have distinguishably different responses to drugs (e.g. Rps27a clones 29 and 44). Beyond cytotoxic conditions (e.g. Latrunculin B at 1.25 mM) few dominating patterns are apparent; neither unperturbed subcellular compartment nor known targets of drugs are major predictors of the degree of perturbation of most experiments.

To assess the learner's ability to find acute changes, we divided experiments into those for which the learner predicted large effects (compared to unperturbed) and those predicted to have small or no change. When we measured the actual effect sizes, the experiments with large predicted effects were indeed observed to have larger effect sizes ($P<0.001$, Mann-Whitney $U$). Furthermore, for those predicted to have an acute change, the predicted *extent* of change was modestly correlated with the observed effect magnitude (Pearson's $r = 0.53$).

One difficulty with this analysis of effect sizes is that the scale of feature distances are likely different for distinct unperturbed patterns (e.g., nucleolar vs. Golgi), a fact not appreciated by us when we began this work. To correct for this, we formed a 48 x 48 matrix of the degrees of perturbation by grouping the clones by visual assessment of the unperturbed phenotypes (i.e., grouping by apparent subcellular pattern). These groups were placed on a common scale by normalizing features within each group by their maximum response. The results are shown in *Figure 5*, in which for display purposes we clustered the drugs using the average perturbation of each group of clones for each drug. This provides a summary of the extent to which each drug affected the spatial distribution of each tagged protein. Beyond the strong perturbations of the cytotoxic drug conditions (drugs between Latrunculin B 1.25 mM and 0.31 mM), the most striking aspect is that nearly all drugs elicited noticeable changes to protein localization patterns in most experiments. These include effects of perturbagens commonly used in cell biology, such as Brefeldin A, on targets other than their intended compartment (e.g. on nucleoplasmic proteins). A two-way ANOVA model fit to these perturbations (taking each drug cluster as a factor, and each group of clones as an independent factor) indicates that these drug and clone groups are all significantly different ($P<0.05$, Tukey's range test, Bonferroni corrected).

To confirm and illustrate one of the top-ranked predictions, we reimaged cells expressing tagged Fa2h with and without various treatments using a spinning disc confocal microscope. The active learner predicted that both cycloheximide and Econazole would affect Fa2h localization, and that they would have different effects. Fa2h has been previously suggested to be localized to the endoplasmic reticulum (*Eckhardt et al., 2005*), and has been observed to also occasionally localize within the nucleus in MDA-MB-231 cells (*Takeda et al., 2013*) although no conclusive evidence has been presented. In our images, we have observed a broad range of phenotypes encompassing these and other localizations. We imaged Fa2h localization over a 4 hr period (from +2h to +6h after treatment) in a manner otherwise closely resembling the active learning experiment protocol (see Materials and methods). Images of treated vs. vehicle treatments were readily classified by logistic regression (85% accuracy, $n = 73$, $P<0.001$, one-sided Binomial test against p = 0.33, the class-proportional null); see *Figure 6A*. Furthermore, cycloheximide and Econazole treatments were also distinguishable above random guessing (68% accuracy, $n = 47$, $P = 0.006$, one-sided Binomial test against p = 0.51, the class-proportional null). Projecting these data onto the two classifiers gives rise to Gaussian distributions for each treatment; consistent with the active learning predictions (and clustering method used), these three conditions have distinguishable distributions. That these classifications are independent of treatment duration suggests that these distributional shifts may be rapidly attained in the first (unmeasured) 2 hr; we do not have statistical power to reject the possibility that they are in steady state from 2–6 hr thereafter. Taken together, these results confirm our predictions of differences as a result of treatment.

As shown in *Figure 6A*, Fa2h condition-dependent alterations to localization are not acute (as measured by our image features), nor did they appear to completely restrict or translocate between well-defined compartments. While our analysis hereto has deliberately avoided the complications and inconsistencies of human-assessed labels, we now wished to provide a description of the Fa2h localization differences in those terms. To do this, we performed an analog of archetypal analysis (*Cutler and Breiman, 1994*) in which we sought to identify a small number (five) of archetypical cells (single-cell fields), each with some human-assessed description of localization pattern, and then described the remainder of the cells as linear combinations of these archetypes; this process is cartooned in *Figure 6B* (see Materials and methods). As shown in *Figure 6C*, these archetypes are not trivially described. Two exhibit both extra- and subnuclear localization (but curiously are not colocalized with nucleoi, and objects are too large to likely be Cajal bodies). Even the secretory localization patterns are unusual: in several cells a nonuniform perinuclear and early ER pattern appears present, without concomitant late ER or Golgi-associated structures, and with vesicles seemingly too small to be microsomes. Without further evidence of the other contents of these structures, these

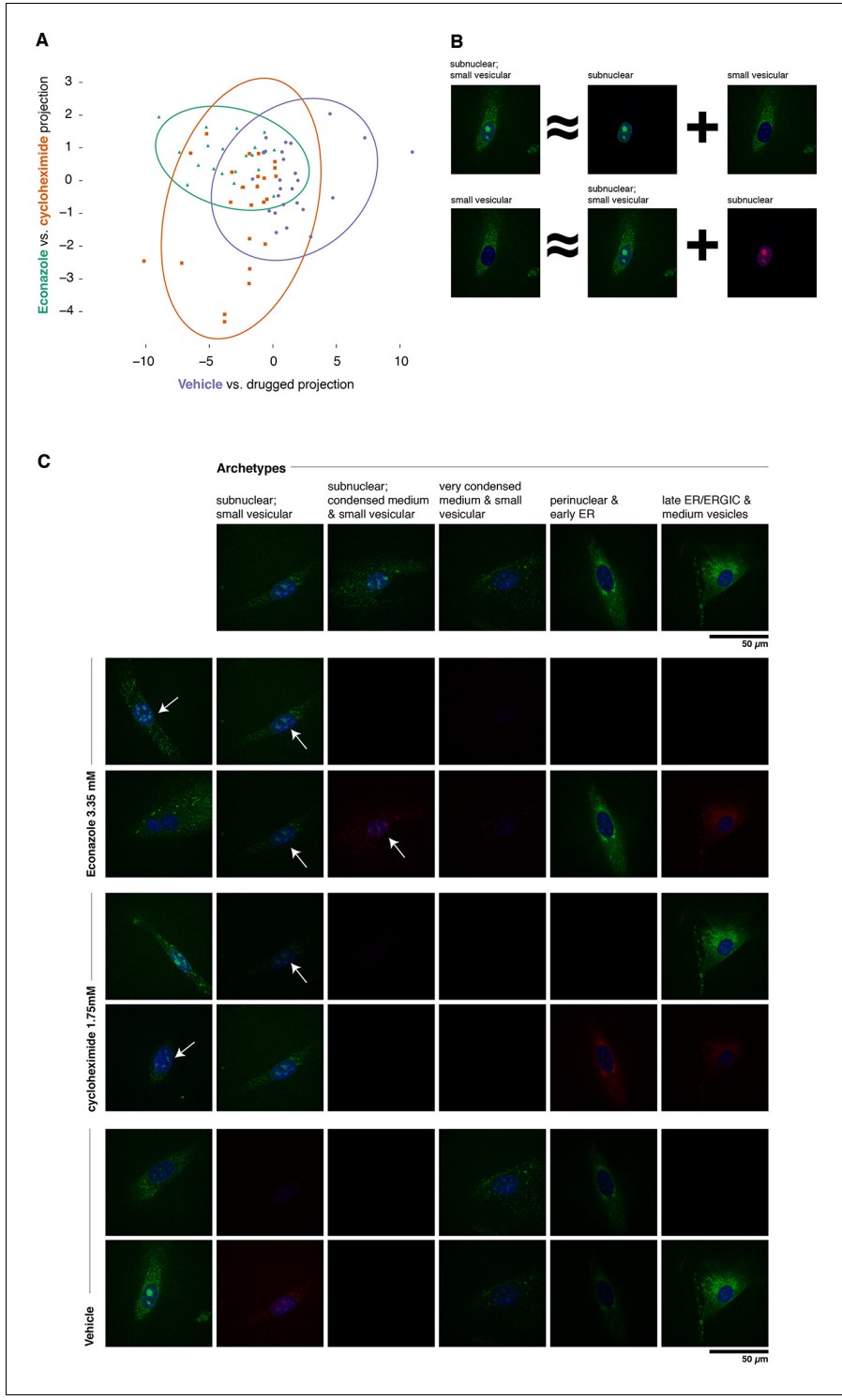

**Figure 6.** Complex phenotypes arising in top-ranked translocations discovered by the active screen. EGFP-tagged Fatty acid 2-hydroxylase (Fa2h, clone number 23) expressed in NIH-3T3 cells exhibits a broad range of localization patterns in two top-ranked treatments, cycloheximide (drug 4), and Econazole (drug 16). (A) Image features calculated from confocal images (60X) of two treatments (orange squares and green triangles, respectively) as well as the vehicle treatment (purple dots) are reasonably well classified by logistic regression. The resulting 2D projection of the 173-dimensional feature space transforms the distribution of each treatment into a 2D Gaussian (95% confidence intervals as colored ellipses). Consistent with the active screen results, drug treatments are distinguishable from vehicle and from each other. Fa2h exhibits an unusual spread of secretory-associated, or

*Figure 6 continued on next page*

*Figure 6 continued*

subnuclear (but not nucleolar), and sometimes both localizations in single cells. (B) In order to visually assess the distributional differences between treatments, we can extend the usual visual vocabulary of coarse localization phenotypes (e.g. 'Golgi,' 'ER' and the like) by decomposing the feature vectors of each single-cell image in terms of a fixed set of single-cell images. Each image can be expressed as an additive combination; for example, a cell exhibiting both a subnuclear and small vesicular localization can be linearly approximated by adding together feature vectors of a subnuclear and a separate small vesicular cell. Subtractive cases (B, bottom row) are false colored with red instead of green for EGFP signal. (C) Archetypal cells (top row), chosen by minimax clustering (*Bien and Tibshirani, 2011*), can be approximately added in different weighted combinations to reveal nuanced differences in condition-dependent localization (see Materials and methods). For each treatment (pair of rows), two cells (leftmost column) and their additive description in terms of the archetypes are displayed. White arrows highlight dim subnuclear signal where visually subtle. Overall, Econazole appears to have the effect of generally enhancing post perinuclear/ER secretory structure localization, whereas cycloheximide generally suppresses ER localization in favor of presumably later secretory vesicular localization.

assessments are of course speculative. Nonetheless, the overall visual decomposition analysis suggests that phenotypic differences are concordant with the known mechanism of action of the two drugs. We emphasize that these issues of phenotype label ambiguity were completely immaterial to the active learner, which merely required that roughly similar images have roughly similar feature vectors.

## Discussion

Thanks largely to the emergence of '-omic' approaches, efficiencies of scale in experimentation have received considerable attention (*Ideker et al., 2001*; *Kitano, 2002*; *Westerhoff and Palsson, 2004*). Our work directly benefits from and leverages modern solutions that reduce the cost of performing a set of experiments: automated microscopy, liquid handling robots for cell culture and compound management, and automated image analysis (*Abraham et al., 2004*; *Boland et al., 1998*; *Macarron et al., 2011*). Our findings extend these benefits into the regime of efficiencies of *scope*.

We have shown that our active learner – previously only characterized through simulation of idealized data – was able to efficiently learn an accurate model of how drugs alter protein subcellular localization in a practical setting. To do this the learner had to overcome several real-world obstacles including extensive variation within and between experiments and the requirement that the set of phenotypes had to be learned as experiments progressed. The results showed not only that an accurate model could be learned in this setting without exhaustive experimentation but that the model could generalize to proteins that it had not observed previously.

The final model contained over 50 clusters, which we have considered to be distinguishable phenotypes. Of these, 31 were observed for more than one clone and drug combination. *Figure 4* allows a visual comparison of the clusters, although it can be very difficult to visually distinguish subcellular patterns that are reproducibly distinguished by numerical features (*Murphy et al., 2003*). Whether these clusters are "biologically relevant" is a difficult question. For example, a drug may cause a slight swelling in an organelle (such as lysosomes or mitochondria) that would be picked up by our numerical features. Is this a unique, biologically relevant phenotype, especially since the extent of the swelling may be difficult to see visually and may depend on the concentration of the drug used? Regardless of what term we use, we can conclude with confidence that that drug affects that organelle (and perhaps that other drugs may have similar effects). To avoid the issue of nomenclature, we can simply consider the clusters to be what they are: changes induced by drugs that were observed frequently enough to be considered significant. Demonstrating the feasibility of automatically and efficiently finding such effects for large sets of drugs and targets was one of the major goals of our study. It is beyond the scope of our study to further characterize each of these changes, but we suggest that a detailed characterization of each of our observed effects be carried out when using any of the drugs we have studied.

Future work on active learning for problems such as ours should account for several methodological shortcomings we identified after completing our experiments. Most importantly, image quality

control was based only on images taken in the first round of experimentation, and so poor quality images of types seen later were not eliminated during the active experimentation. Our concern was that human quality control might bias the experimentation (e.g. subtle phenotypes not obvious to a human might be missed); however our results, particularly phenotype identification, would have likely been improved had we allowed ourselves to monitor the data being collected. Our results also suggest that future active learning projects of this type should directly assess differences in the variation in measurement results for different experiments (i.e., different drug-target combinations), which may require making decisions (i.e., during acquisition) about which combinations require additional image collection.

Given the small size of the experiment space that our budget allowed us to study, we cannot claim that our findings generalize to all proteins or all possible perturbagens. The results suggest that the model is not simply learning a solution to the problem given (that of 96 clones and 96 drugs) and that these predictions are well correlated to underlying biological pathways represented here by subcellular localization. Future experiments exploiting this generalization of the learned model could result in accurate predictions for novel drug/target pairs to further reduce the cost of discovery for potential drug/target interactions. This is a major benefit of our approach compared to the typical single-target based approaches that are widely used today.

## Materials and methods

### Availability
All images collected and software used for this work will be made available upon publication at http://murphylab.web.cmu.edu/software.

### Clone generation, storage and preparation
Clones expressing different EGFP-tagged proteins were generated as previously described (*Coelho, 2013*; *Garcia Osuna et al., 2007*). The tagged gene was identified by sequencing the region near the retroviral insertion (*Kangas et al., 2016*). 48 clones were chosen; see *Supplementary file 1*. To ensure consistency of tagged protein expression, localization and drug responsiveness across rounds of imaging, clones were grown to large quantities in cell culture factories (Nunc) per manufacturer instructions and stored in 1 mL cryotubes (Corning) at -80°C in freezing media as described elsewhere (*Hay, 1978*). All clones were kept constantly available by periodic replenishment from frozen stocks. Dishes were grown to ~90% confluence and used within four passages of thawing.

### Compound library storage and preparation
Compounds (*Table 1*) were solubilized once in pure DMSO. 1.3 µl of each was individually pipetted with the liquid handling robot into many Eppendorf tubes and stored at -4°C until needed. When requested by the learner for an experiment a tube was taken out and placed in the liquid handling robot. Room temperature imaging media (1 ml Opti-MEM (Invitrogen, Carlsbad, CA) supplemented with 1.25 µg/ml Hoechst 33,342 (Invitrogen)) was pipetted by the robot directly into the tube to produce the media used both for drug treatment and imaging. This procedure was adopted to control pipetting and dilution inaccuracies, and to form the final solution in one step. From these tubes, a 384-well microtiter plate (Nunc) was filled in correspondence to the experiment arrangement.

### Imaging plate design
Clones were plated for imaging as follows: the contents of a 60 mm dish were trypsinized (0.1% v/v) (Invitrogen), counted with a hemacytometer, resuspended in growth media, and ~250,000 cells were pipetted into a 1.5 ml Eppendorf tube. These Eppendorf tubes were placed in a liquid handling robot (Eppendorf, Hauppauge, NY) and plated under the control of a Python script that generated commands for the robot. Each experiment well received ~6250 cells. Technical triplicates were randomly arranged on the plate. The 96 x 3 experiment wells were augmented by 20 wells reserved for imaging controls (a subset of the clones from the 48), also randomly placed. After allowing the cells to attach for 24 hr, the medium was removed and replaced by the drug-containing media from the

384-well drug microtiter plate using the liquid handling robot. Drug additions were timed so that the period between the first and last wells was 6 hr.

## Microscopy

All microscopy was performed using an automated microscope, the IC100 (originally manufactured by Beckman-Coulter and maintained by Vala Sciences, San Diego, CA) equipped with a Nikon S Fluor 40x/0.9 NA objective. Identical settings were used across all plates; in particular, camera gain for Hoechst emission was set to 0.37, with a 16.6 ms integration time, camera gain for EGFP emission was set to 0.6, with a 4 s integration time. Mercury arc bulbs were exchanged after as close to 100 hr as possible to attempt to control illumination variation. The pixel size in the sample plane was 0.161 x 0.161 μm. For confirmatory experiments, an Andor (Concord, MA) Revolution XD System spinning disk confocal microscope with a Nikon Plan Fluor 60x/1.4 NA objective and a pinhole diameter of 50 μm was used. The pixel size in the sample plane was 0.174 x 0.174 μm.

The images for each experiment in the 96 x 96 space were kept separate, i.e., the images were stored under the number of the clone and drug that the learner had requested and not mixed with images for the other numbers of that clone and drug. Thus if the learner requested an experiment for clone and drug numbers that corresponded to the same actual clone and drug as a previously performed experiment, new images were collected rather than providing the existing images so that the learner could not detect the correspondence between clones or drugs by exact matching of images.

## Image analysis and quality control

Images were individually contrast stretched and represented by SLF34 feature vectors (one per image) as described previously (*Coelho et al., 2010*). These whole image ("field level") feature vectors describe the fluorescence patterns of protein and DNA stains relative to each other and are not directly interpretable by humans as fixed subcellular locations. Automatic quality control was applied to filter out poor quality images: e.g., fields that contained no cells, were overly confluent, or were out of focus. This was done using a random forest classifier (*Breiman, 2001*) that was trained before the beginning of the active learning experiments on data collected for the initial 96 (no-drug) experiments from human labeling of 600 fields. If no images for an experiment passed quality control, the experiment was considered as not having been performed. From raw data each round, for the subset passing quality control, the collected features were column centered (i.e. set to zero mean and unit standard deviation) and a subset of linearly independent features were identified (by the Gram-Schmidt process) and used for subsequent analysis.

## Phenotype determination by clustering

A form of agglomerative hierarchical clustering was used. The leaves of the cluster tree corresponded to experiments; each experiment was associated with the set of feature vectors for the images obtained for it, which varied in size depending on how many images passed quality control. A single-linkage clustering over sets of experiments (as opposed to individual feature vectors) was performed; internal nodes of the tree were associated with the union of the set of feature vectors of their descendants. At every level of the tree, a score was computed between each pair of nodes. This score ranged from one (1) ("totally intermingled point sets") to zero (0) ("totally dissimilar point sets"). The score for nodes *A, B* was computed by measuring the average performance of a 1-nearest neighbor classifier (between *A, B*) taken over of five (5) independent draws (with replacement), with at most 500 points chosen from each node, and equal numbers of points from each node. The pair of nodes with the greatest score (ties arbitrarily broken) were merged to form a new node in the next level of the tree. The pairwise scores had to be recomputed at each level in the tree for the new node corresponding to the merged nodes; scores for pairs of nodes that were not merged were unaffected. The merging process terminated with a single node, associated with the complete data. The cutoff for the clustering tree was determined relative to five (5) independent data splitting samples. That is, for each data splitting, each experiment was divided equally, randomly and disjointly into two leaves; these were assumed to be equivalent distributions. The goal was to identify a threshold such that experiments at least as similar as the variation within an experiment were clustered together. For each of these data splits, the cutoff for the original clustering tree was set to the

average (across data-splittings) score of the least level (greatest number of clusters) where at least 90% of the experiment data splits were coclustered. The original clustering tree was pruned using this cutoff to identify clusters (phenotypes).

## Active learning experimentation

In brief, the learning process initialized with observations of each clone under a no-drug condition in technical triplicates. Each round of learning consisted of the following steps. First, the data passing quality control were clustered as described above to form phenotypes. A list of the experiments and their phenotypes were given to the active learner software (written in reFLect [*Grundy et al., 2006*] and described previously [*Naik et al., 2013*]). To the active learner, experiments were abstractions corresponding to 'Target,' 'Condition' – no specific information about these (e.g. their true identities, chemical composition or amino acid sequence, etc.) were used. The active learner used these data make a predictive model, which was formed in stages by the following process. The "inductive bias" of the model is that there are fewer distinct Target-types than targets (i.e. there are likely proteins that exhibit the same phenotypes under similar conditions), and that there are fewer distinct Condition-types than conditions (by similar reasoning). Each Target-type was a set of targets such that they had the same measured phenotype for all conditions they were both measured in. In turn, a Target-type predicted that all of its targets had the same phenotype for all conditions, even if they had not been measured yet. Each target was in one Target-type. Condition-types were similar and constructed using Target-types: between any two conditions in a Condition-type, Target-types did not have different phenotype predictions. Condition-types were identified in one step by solving a constrained logic program to find the smallest overall number of Condition-types, and Target-types were identified by a greedy, iterative pairwise merging approach which first collapsed the greatest overlapping conditions first, and then opportunistically compressed Condition-types if they were disjoint. Predictions – if they existed – were a matter of looking up the corresponding Target- and Condition-type for a given target and condition. The active learner used this model to select experiments by the following procedure. Overall, target and condition combinations that the model did not have a prediction for were prioritized, and then the rest were treated equally. Repeatedly, until 96 new experiments were chosen, a random high-priority experiment was chosen; all other unobserved experiments in the same Target- and Condition-type as that experiment were temporarily removed from consideration. If the remaining set was exhausted before selecting 96 new experiments, those unobserved experiments which had been removed from consideration were put back into consideration, and the same process was applied again. In this way, the set of experiments overall maximized the diversity of Target- and Condition-types, which presumably would best test their predictions, given new data. Models were learned anew each round, and so the only 'history' the active learner was made aware of was the aggregate effect of having selected some experiments previously, but not their ordering.

## Accuracy assessment by classification of predictions

The model at each round was based on observing a subset of the total set of experiments. The images corresponding to each observed experiment were associated with a phenotype by clustering of their SLF34 features, as described above. Each model also made phenotype predictions for thereto unobserved experiments, data for which were collected later. For each feature vector $U$ corresponding to an image from an unobserved experiment, we determined which observed image $O$ was closest to it (in the Euclidean metric). We then associated that feature vector $U$ with the phenotype assigned to $O$. This is the nearest-neighbor classification of the unobserved data. These classifications were used to determine, for each unobserved experiment, if the phenotype with the plurality of classifications matched the model prediction.

## Model generalization to novel clones

The six clones not in the 96 x 96 experiment space, which were never used during active learning, allow us to make an independent assessment of how well the model generalizes predictions to unseen and/or novel targets. To do this we collected images and calculated features for the full 6 x 48 space of these six clones for all drugs. Given just the features for the no-drug condition, we matched each novel clone to the clone (or group of clones) from the 48 x 48 space whose no-drug

features were closest. We then predicted phenotypes for each drug for each novel clone (using the model for the matching clone(s) from the last active learning round) and compared them to the measured features as described for unobserved experiments.

## Minimum number of experiments required in an ideal case

Consider an experimental space consisting of *n* drugs and *n* clones. Assume the $n^2$ experiments have distinct phenotypes. Organize the (2*n*, 2*n*) (duplicated) experiment space by indexing drugs and clones in repetition in a matrix; that is the sequence 1,..,*n*,1..*n* is represented as 1,..,*n*,*n*+1,..2*n*. By observing the upper left submatrix (1..*n* x 1..*n*), all of the phenotypes have been observed. Observing the diagonal (1,..,*n* x *n*+1,..2*n*) enables determining the identity of drugs *n*+1,..2*n*. Similarly, observing the diagonal (*n*+1,..2*n* x 1,..,*n*) enables determining the identity of clones *n*+1,..2*n*. If one knew that the drugs and clones were duplicated but did not know which duplicates corresponded to each other, one would only need to sample $n^2+2n$ out of $4n^2$ experiments to obtain a model with perfect accuracy. Note that our active learner did not know in advance that there was any duplication.

## Estimation of random sampling performance

1,000 simulations of random experiment selection were performed. In each, matching the number of experiments the active learner observed per round, experiments were chosen without replacement from the 96 x 96 experiment space. In each simulation, and at each simulated round, histograms of quad coverages were computed, from which an estimated accuracy was calculated by using the regression coefficients identified from the actively learned data. The proxy for random learning accuracy was constructed by taking average of these 1000 simulations per-round.

## Estimation of non-randomness of sampling of quads observed in active learning

Denoting the coefficient of the monomial of the umbral variable *x* to the *n*th power as $[x^n]$ we can use generating functions to compute the number of ways of throwing *z* many indistinguishable balls into up to *b* many distinguishable bins so as to hit *f* many bins, each with a capacity of up to 4 balls as:

$$\frac{\binom{b}{f}[x^z](x+x^2+x^3+x^4)^f}{[x^z](1+x+x^2+x^3+x^4)^b}$$

This can be understood as counting the number of ways of not hitting (b-f) many bins out of b, and then putting at least one ball into f many bins with a total of z throws. The total number of ways of throwing balls without fill constraints is the denominator for this case of distinguishable bins (bins are the experiments, which are distinguishable). We then apply the formula above to compute a p-value for the chance that the coverage of quads observed for the actively learner could have been achieved at random. For the last actively learned model, there were *z* = 2697 'balls' thrown into *b* = 2304 bins. Since that model covered *f* = 1670, we then sum the probabilities when covering *f* = 1670..2304 bins to compute the probability that at least 1670 bins would have been hit by a random process. We computed this with the aid of *Mathematica*.

## Posthoc image quality control

SURF (*Bay et al., 2006*) features were calculated for each image using just the GFP channel, restricting the interest points to be within ~10 μm (150 pixels) of a segmented nucleus. The distributions of these interest point features per image were the atoms of classification in a nearest neighbor two-class classifier (whether or not an image was out-of-focus or contained artifacts), where inter-atom distances corresponded to a kernelized two-sample test as described elsewhere (*Gretton et al., 2012*). To label these data, repeated and nearly exhaustive manual annotation over many iterations were performed.

## Confirmatory Fa2h localization analyses

Fa2h-tagged cells were plated at the same density as for the active learning study, with the exception of being plated in 96-well plates (Nunc) in order to accommodate the confocal microscope. The same previously generated drug aliquots from stock were used to match the active learning conditions as closely as possible. The automated microscope used in the active learning study did not align image fields to center cells, and so to simulate comparable imaging conditions (and any field level feature artifacts) no attempt was made to center cells in fields or to adjust imaging settings (0.4 s and 0.8 s exposure for 440 and 488 nm, fixed gain at 300 (arb. units)). Five (5) fields were taken per well. Sequential wells cycled through each of the three treatments (drugs 16, 4, and 48). 106 fields were acquired over 4 hr, a period starting from +2 hr after drug addition, through the +5 hr timepoint used for the active screen, to +6 hr. 33 fields were discarded for being low contrast, generally occurring at time points immediately after laser and microscope restarts due to hardware and software faults. SLF34 features were calculated per field as before, and Gram-Schmidt process feature selection was used to select 71 features for further use. Classification was by three-fold cross-validated L2-penalized logistic classification and used all 73 fields passing quality control (1–2 cells or cell fragments/field). Archetypical cells were chosen as the centers of the minimax hierarchical clustering (*Bien and Tibshirani, 2011*) of the SLF34 features of each image containing one cell; the highest level of the cluster tree containing 5 nontrivial (nonsingular) clusters was used. Archetype decompositions of the other fields (including polynucliated and multiple cell fields) was calculated by Lasso regression by Mairal's method (*Mairal, 2013*) with the penalization term (set to 1.0) chosen heuristically to force sparse decompositions.

## Acknowledgements

This work was supported in part by NIH grants R01 GM075205, T32 EB009403-01 and P41 GM103712.

## Additional information

### Funding

| Funder | Grant reference number | Author |
| --- | --- | --- |
| National Institutes of Health | GM075205 | Robert F Murphy |
| National Institutes of Health | EB009403 | Armaghan W Naik |
| National Institutes of Health | GM103712 | Robert F Murphy |

The funders had no role in study design, data collection and interpretation, or the decision to submit the work for publication.

### Author contributions

AWN, Conception and design, Acquisition of data, Analysis and interpretation of data, Drafting or revising the article; JDK, Conception and design, Acquisition of data, Analysis and interpretation of data; DPS, Acquisition of data, Analysis and interpretation of data, Drafting or revising the article; RFM, Conception and design, Analysis and interpretation of data, Drafting or revising the article

### Author ORCIDs

Armaghan W Naik, http://orcid.org/0000-0002-7844-2832
Joshua D Kangas, http://orcid.org/0000-0002-6742-5113
Devin P Sullivan, http://orcid.org/0000-0001-6176-108X
Robert F Murphy, http://orcid.org/0000-0003-0358-901X

## Additional files

### Supplementary files

• Supplementary file 1. This spreadsheet contains the RandTag clone names, tagged gene and sub-cellular location annotations for the clones used in this work.

• Supplementary file 2. The spreadsheet contains the average feature values for all measured experiments, the round each was measured, and the cluster number they were assigned to in each round.

• Supplementary file 3. The spreadsheet contains the subcellular pattern labels assigned to each group of phenotypes in *Figure 4*. An image classifier was trained using the pattern labels of *Figure 5* and applied to those images in each group in *Figure 4*. The label assigned with the highest frequency is shown, and any additional labels that were assigned with a probability greater than 15% are also shown.

• Supplementary file 4. The spreadsheet contains the average feature values for the images for all experiments that passed the post-hoc image quality filtering process, the class each clone was assigned to by visual inspection (for the unperturbed condition), and the distance values that give rise to *Figure 5*.

### Major datasets

The following datasets were generated:

| Author(s) | Year | Dataset title | Dataset URL | Database, license, and accessibility information |
|---|---|---|---|---|
| Armaghan W Naik, Joshua D Kangas, Devin P Sullivan, Robert F Murphy | 2016 | RandTagAL: Fluorescence microscope images collected as specified by active learner | http://murphylab.web.cmu.edu/data/RandTagAL.html | Licensed under a Creative Commons Attribution-NonCommercial-ShareAlike 4.0 International License. |

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
