## [Decision Letter]

Thank you for submitting your work entitled "Active Machine Learning-driven Experimentation to Determine Compound Effects on Protein Patterns" for peer review at *eLife*. Your submission has been favorably evaluated by Aviv Regev (Senior editor), Uwe Ohler (Reviewing editor), and three reviewers.

The reviewers have discussed the reviews with one another and the Reviewing editor has drafted this decision to help you prepare a revised submission.

Summary:

The reviewers all agreed with the premise of the manuscript, that there is a need to integrate laboratory automation and active learning to speed up the generation of biological knowledge. Rather than the traditional approach of trying to infer cellular mechanisms, the authors suggest that we turn the problem over to machine learning to choose experiments based on sound statistics. The manuscript thus has the potential to be a valuable contribution highlighting the potential of machine learning for automated experimental design. The authors convincingly demonstrate that active learning improved prediction performance, and one reviewer was impressed by the significant technical achievement of physically implementing 30 rounds of active learning.

The paper should be of interest to a broad readership and could potentially be of significant impact. However, all reviewers also agreed that the structure, description and presentation need to be greatly improved to make the paper reasonably self-contained; in its current state, it was simply too difficult to follow.

The strongest need was perceived for the explanation of the methods: The description of the machine learning methodology is completely absent; the reader is largely referred to the previous paper of Naik et al. (2013). Given the central importance of machine learning in the manuscript, sufficient description of the methodology should be included. The authors need to greatly clarify their approach and rationale, including specific examples of what they are trying to do.

The precise definition of the problem and criteria how to evaluate are unclear in several places.

In turn, the Results section could convey the same amount of information in far less space, and the Discussion section was seen as bloated.

We provide below the essential revisions.

Essential revisions:

Specific condensed comments from reviewer one:

1.1) Are images in a quad actually the same image (as seems to be implied in the subsection “Efficiency of Learning”), or do they correspond to biological replicates (as seems to be the in the subsection “Identifying Perturbations”)? Please clarify.

1.2) Is the number of clusters fixed at the outset? Could it be varied when e.g. new treatments result in unexpected patterns? This would be worth discussing as one of the strengths of the approach is the absence of a need for defining the number of classes.

1.3) In the fourth paragraph of the subsection “Efficiency of Learning”: Explain more explicitly the regression model proposed so that we understand what the regression coefficients mean exactly.

1.4) In the second paragraph of the subsection “Robustness of Learning to Imperfect Phenotype Identification”: the discussion of confused quads is quite confusing.

1.5) In the second paragraph of the subsection “Identifying Perturbations”, the assessment of prediction of effect is convoluted, first discretising, and then evaluating an auROC. Why not directly regress/correlate real effect magnitude with predicted effect magnitude?

1.6) Earlier examples of active learning in a biological context should be referenced, e.g. Romero, Krause and Arnold, PNAS 110, no. 3, 2013.

Specific condensed comments from reviewer three:

2.1) The basic definition of a "correct" prediction is obscure:

"We defined correctness of a predicted phenotype for an experiment to be when the plurality of observations for that experiment is most similar to the examples the learner used to construct that phenotype (see Materials and methods)."

The use of the word "plurality" is unclear. It sounds like the correctness is defined *relative* to the training data, which seems very unlikely to generalize as new phenotypes are included.

The authors should give a specific example of what a prediction looks like (something like a subcellular localization class? or set of classes? or feature vector?) along with an unseen observation and explain how they decide if the prediction is accurate or not.

There is a Methods section entitled "Accuracy Assessment by Classification of Predictions", which has some discussion of nearest-neighbour classification, but it is unclear how the "correct" vs. "incorrect" decision is made.

2.2) Apparently, the authors duplicate their data, but hide this from the learning algorithm.

"The goal of the duplication was to provide some guaranteed basis for the learner to be able to predict at least some results without performing all possible experiments."

"From the design of the study, each unique combination of drug and clone corresponds to four potential 19 experiments in the 96x96 space (which we refer to as a quad)."

However, the claim in the Abstract is:

"The results represent the first practical demonstration of the utility of active learning-driven biological experimentation in which the set of possible phenotypes to be learned is unknown in advance."

This seems contradicted by the "hidden" duplication structure of the data, and is made unclear by the terminology ("quad"). Is the "duplication" scientific "replication" (in the sense of doing the same experiment twice)? Are the same clones reimaged (technical replicates)? Is the experiment performed independently (biological replicates)?

If there are replicates, why not use this to improve the statistical analysis, or hold out some of the replicate data to evaluate the accuracy of the approach? For example, why not run the learner on two replicates independently and see how well the results agree.

The authors should run the active learner in a practical scenario, and then evaluate the accuracy of the findings. The real test of the methodology may be the extent and efficiency with which the active learner discovers the most interesting unknown drug effects on phenotypes. Presumably very rare or subtle effects are harder to find and more interesting, and if the learner can find these, it must be doing well. This is not convincingly demonstrated (or explained) in the paper.

2.3) It is unclear why the accuracy is not evaluated on the held out data:

"Assuming that the same accuracy per coverage model holds for random samples (that is, that the accuracy of the model can be accurately predicted from just the distribution of quad samplings)".

The authors report that the number of phenotypes increases with more data. How would the accuracy stay the same as more data is collected (especially if the accuracy is evaluated relative to the data seen so far)?

2.4) "These distances were then thresholded by fitting a two-class Gaussian mixture model which 10 set ~25% of the experiments as significantly perturbed. Perturbation of model predicted phenotypes was defined similarly by pooling data within phenotypes instead. Using these we constructed a receiver operator curve; the area under this curve (AUC) was 0.68, which suggests overall agreement in what may reasonably be considered significantly perturbed experiments in a post-hoc analysis."

In this evaluation, the authors define a clear measure of accuracy. However, this seems like a different problem than the one they set out to answer. The Abstract states:

"To our knowledge this is the first series of active learning-driven prospective biological experiments where the possible answers (e.g., what phenotypes might be observed) were not"

This sounds as if the general aim is to predict the actual phenotypes (or at least phenotype classes) – but here the question is simply whether the image is different from vehicle. This alone would be a very interesting (and difficult) problem, and it would be fine to center the paper around these results.

2.5) "The generally increasing number of phenotypes the model identified as more data were collected (Figure 3)."

An exciting aspect of this work is the application in situations where the phenotypes were unknown in advance. However, there is no evaluation of how well the learner is doing at recognizing the localization phenotypes. Figure 3 shows that there might be as many as 50 phenotypes. What are these phenotypes? Are they biologically relevant? Imaging artefacts? Overfitting the data?

2.6) "To confirm and illustrate one of the top-ranked predictions…"

What was the precise prediction – that Fa2H was localized to the ER-Golgi? That cyclohexamide and econazole have opposite effects? Or that the drugs would have effects at all? Either way, the authors should show more than one cell in the panels and give statistical measurements of the patterns. As it stands, it is unclear what Figure 5 is meant to demonstrate.

[Editors' note: further revisions were requested prior to acceptance, as described below.]

Thank you for resubmitting your work entitled "Active Machine Learning-driven Experimentation to Determine Compound Effects on Protein Patterns" for further consideration at *eLife*. Your revised article has been favorably evaluated by Aviv Regev (Senior editor), a Reviewing editor, and one reviewer.

The manuscript has been substantially improved but there are some remaining issues that need to be addressed before acceptance. Please look at the comments from the remaining reviewer and respond and revise your paper to clarify the two points raised. We emphasize (1) that the paper needs to be written in a manner accessible to the *eLife* readership of biologists; this can be done with better writing, and is imperative; (2) the authors cannot wave away the possibility of technical artifacts; and (3) proper validation, with appropriate statistics should be addressed in the new phenotype.

*Reviewer #3:*

The authors have done a great job improving the clarity of their paper. However, I still believe that it will be very difficult for a general biology audience to understand. The authors still use a lot of non-standard terminology and convoluted exposition to describe their work.

The authors addressed most of my comments, although I was not satisfied with their responses to two major points:

1) In my first comments, I wrote:

Figure 3 shows that there might be as many as 50 phenotypes. What are these phenotypes? Are they biologically relevant? Imaging artefacts? Overfitting the data?

The authors answered:

“We thank the reviewer for raising this useful point. We have clarified our goal regarding phenotypes in the Introduction, and have extensively revised the section "Identifying Perturbations." We believe that the most direct answer is that it is reasonable to consider as "biologically relevant" phenotypes that are statistically significant and when predicted to be observed for as yet untested combinations of drugs and targets match subsequently corrected images.”

I strongly disagree with the authors on this point. In my experience, technical artifacts typically show the strongest statistical significance in automated microscopy image analysis. The authors should show convincing examples of the phenotypes they believe are "biologically relevant" or remove the claims that their approach can identify new phenotypes.

2) I also asked for clarification and improvement of their validation of a new phenotype in Figure 5. The authors have done a good job to clarify the explanation of their independent test of a prediction in Figure 5. However, they still only show one cell in each panel, provide no statistical evidence that these patterns are actually different from the vehicle, nor do they show that the changes observed in these new images are consistent with the images that were used by the active learner. I therefore find this "validation" unconvincing.

---

## [Author Response]

The reviewers all agreed with the premise of the manuscript, that there is a need to integrate laboratory automation and active learning to speed up the generation of biological knowledge. Rather than the traditional approach of trying to infer cellular mechanisms, the authors suggest that we turn the problem over to machine learning to choose experiments based on sound statistics. The manuscript thus has the potential to be a valuable contribution highlighting the potential of machine learning for automated experimental design. The authors convincingly demonstrate that active learning improved prediction performance, and one reviewer was impressed by the significant technical achievement of physically implementing 30 rounds of active learning.

*The paper should be of interest to a broad readership and could potentially be of significant impact. However, all reviewers also agreed that the structure, description and presentation need to be greatly improved to make the paper reasonably self-contained; in its current state, it was simply too difficult to follow. The strongest need was perceived for the explanation of the methods: The description of the machine learning methodology is completely absent; the reader is largely referred to the previous paper of Naik et al. (2013). Given the central importance of machine learning in the manuscript, sufficient description of the methodology should be included. The authors need to greatly clarify their approach and rationale, including specific examples of what they are trying to do. The precise definition of the problem and criteria how to evaluate are unclear in several places. In turn, the* Results section *could convey the same amount of information in far less space, and the* Discussion section

*was seen as bloated. We provide below the essential revisions.*

As requested, we have included a detailed description of the active learner itself as in the Methods section “Active Learning Experimentation” and have also given a brief description of the learner in the Results section “Experiment Space Construction and Active Learning.” We have significantly reorganized the Results section (including removing unnecessary analyses) and shortened the Discussion.

Essential revisions: Specific condensed comments from reviewer one: 1.1) Are images in a quad actually the same image (as seems to be implied in the subsection “Efficiency of Learning”), or do they correspond to biological replicates (as seems to be the in the subsection “Identifying Perturbations”)? Please clarify.

We thank the reviewer for this observation and have clarified in the first paragraph of the Results and in the section on “Efficiency of Learning.” The images in a quad were *logically* biological replicates but were presented to the active learner completely separately.

*1.2) Is the number of clusters fixed at the outset? Could it be varied when e.g. new treatments result in unexpected patterns? This would be worth discussing as one of the strengths of the approach is the absence of a need for defining the number of classes.*

The number of clusters was never fixed, and was determined in each round as new experiments were performed. We have clarified this in the second paragraph of the Results and this is also mentioned in the Discussion.

*1.3) In the fourth paragraph of the subsection “Efficiency of Learning”: Explain more explicitly the regression model proposed so that we understand what the regression coefficients mean exactly.*

We thank the reviewers for raising this, and have substantially extended and revised the original description to aid the reader in interpreting these results (second paragraph of section “Efficiency of Learning”).

*1.4) In the second paragraph of the subsection “Robustness of Learning to Imperfect Phenotype Identification”: the discussion of confused quads is quite confusing.*

We agree with the reviewers. We have removed those analyses for two reasons: they are not strictly necessary for the perturbation analysis we present at the end of the manuscript, and they disturbed the logical flow.

*1.5) In the second paragraph of the subsection “Identifying Perturbations”, the assessment of prediction of effect is convoluted, first discretising, and then evaluating an auROC. Why not directly regress/correlate real effect magnitude with predicted effect magnitude?*

We thank the reviewer for this excellent suggestion and have replaced the original analysis with the reviewer’s suggestion in the second paragraph of the section “Identifying Perturbations.”

*1.6) Earlier examples of active learning in a biological context should be referenced, e.g. Romero, Krause and Arnold, PNAS 110, no. 3, 2013.*

We have added this reference.

*Specific condensed comments from reviewer three: 2.1) The basic definition of a "correct" prediction is obscure: "We defined correctness of a predicted phenotype for an experiment to be when the plurality of observations for that experiment is most similar to the examples the learner used to construct that phenotype (see Materials and methods)." The use of the word "plurality" is unclear. It sounds like the correctness is defined relative to the training data, which seems very unlikely to generalize as new phenotypes are included.*

*The authors should give a specific example of what a prediction looks like (something like a subcellular localization class? or set of classes? or feature vector?) along with an unseen observation and explain how they decide if the prediction is accurate or not.*

*There is a Methods section entitled "Accuracy Assessment by Classification of Predictions", which has some discussion of nearest-neighbour classification, but it is unclear how the "correct" vs. "incorrect" decision is made.*

We have extensively clarified the accuracy assessment. We split the accuracy discussion into a separate section. The procedure and rationale was itself newly elaborated in the section “Accuracy of Learning”.

*2.2) Apparently, the authors duplicate their data, but hide this from the learning algorithm. […]*

*The authors should run the active learner in a practical scenario, and then evaluate the accuracy of the findings. The real test of the methodology may be the extent and efficiency with which the active learner discovers the most interesting unknown drug effects on phenotypes. Presumably very rare or subtle effects are harder to find and more interesting, and if the learner can find these, it must be doing well. This is not convincingly demonstrated (or explained) in the paper.*

We thank the reviewer for raising this, and have revised the manuscript to clarify and elaborate. We explain more clearly the rationale behind the duplication at the beginning of the Results section. We also emphasize the nature and identify of held out data both within the 96x96 space (first paragraph of “Accuracy of Learning”), and for the entirely held out experiments in the 48x6 space (second paragraph of that section). Please note that we have added a way of estimating the improvement of accuracy over random for the 48x6 space and in doing that we noticed an error in the way accuracy was being calculated for these – we had erroneously analyzed the 48x6 data with the wrong round’s model/phenotypes (round 29 rather than round 30). In view of this, we rechecked and reran all of the accuracy calculations and verified that they are all correct.

Regarding the “quad” terminology, we have clarified its definition at the beginning of the “Accuracy of Learning” section and alluded to it in the section “Experiment Space Construction and Active Learning”.

*2.3) It is unclear why the accuracy is not evaluated on the held out data: "Assuming that the same accuracy per coverage model holds for random samples (that is, that the accuracy of the model can be accurately predicted from just the distribution of quad samplings)". The authors report that the number of phenotypes increases with more data. How would the accuracy stay the same as more data is collected (especially if the accuracy is evaluated relative to the data seen so far)?*

We understand that this was not explained clearly and have clarified as discussed for the previous point. In addition, we further clarified this as the beginning of the section “Estimated Accuracy from Random Learning.”

*2.4) "These distances were then thresholded by fitting a two-class Gaussian mixture model which 10 set ~25% of the experiments as significantly perturbed. Perturbation of model predicted phenotypes was defined similarly by pooling data within phenotypes instead. Using these we constructed a receiver operator curve; the area under this curve (AUC) was 0.68, which suggests overall agreement in what may reasonably be considered significantly perturbed experiments in a post-hoc analysis." In this evaluation, the authors define a clear measure of accuracy. However, this seems like a different problem than the one they set out to answer. The Abstract states: "To our knowledge this is the first series of active learning-driven prospective biological experiments where the possible answers (e.g., what phenotypes might be observed) were not" This sounds as if the general aim is to predict the actual phenotypes (or at least phenotype classes)* –

*but here the question is simply whether the image is different from vehicle. This alone would be a very interesting (and difficult) problem, and it would be fine to center the paper around these results.*

We thank the reviewer for this feedback and have clarified at the beginning of the section “Identifying Perturbations.” As discussed above under point 1.5, we have removed the AUC analysis and replaced it with estimation of the correlation between predicted and observed extent of perturbation.

*2.5) "The generally increasing number of phenotypes the model identified as more data were collected (Figure 3)." An exciting aspect of this work is the application in situations where the phenotypes were unknown in advance. However, there is no evaluation of how well the learner is doing at recognizing the localization phenotypes. Figure 3 shows that there might be as many as 50 phenotypes. What are these phenotypes? Are they biologically relevant? Imaging artefacts? Overfitting the data?*

We thank the reviewer for raising this useful point. We have clarified our goal regarding phenotypes in the Introduction, and have extensively revised the section “Identifying Perturbations.” We believe that the most direct answer is that it is reasonable to consider as “biologically relevant” phenotypes that are statistically significant and when predicted to be observed for as yet untested combinations of drugs and targets match subsequently corrected images. We have added a comment to this effect to the Discussion.

*2.6) "To confirm and illustrate one of the top-ranked predictions…" What was the precise prediction* –

*that Fa2H was localized to the ER-Golgi? That cyclohexamide and econazole have opposite effects? Or that the drugs would have effects at all? Either way, the authors should show more than one cell in the panels and give statistical measurements of the patterns. As it stands, it is unclear what Figure 5 is meant to demonstrate.*

We have clarified in the Introduction that our goal is not to assign terms to the patterns or their perturbations. We also added a statement of the prediction that the reimaging was to confirm: two different effects. The reimaging was done with confocal microscopy was done to give higher resolution for visualization of what appear visually as subtle changes in the original images. The statistical measurements of the patterns in these images were the basis for the assignment of the drugs to different phenotypes.

[Editors' note: further revisions were requested prior to acceptance, as described below.]

*The manuscript has been substantially improved but there are some remaining issues that need to be addressed before acceptance. Please look at the comments from the remaining reviewer and respond and revise your paper to clarify the two points raised. We emphasize (1) that the paper needs to be written in a manner accessible to the eLife readership of biologists; this can be done with better writing, and is imperative; (2) the authors cannot wave away the possibility of technical artifacts; and (3) proper validation, with appropriate statistics should be addressed in the new phenotype. Reviewer #3: The authors have done a great job improving the clarity of their paper. However, I still believe that it will be very difficult for a general biology audience to understand. The authors still use a lot of non-standard terminology and convoluted exposition to describe their work.*

We understand this concern and have revised the manuscript again in a number of places to further improve the accessibility for biologists. We especially modified the Introduction and the Results sections on Experiment Space Construction and Active Learning and Efficiency of Learning.

*The authors addressed most of my comments, although I was not satisfied with their responses to two major points: 1) In my first comments, I wrote: Figure 3 shows that there might be as many as 50 phenotypes. What are these phenotypes? Are they biologically relevant? Imaging artefacts? Overfitting the data? The authors answered: “We thank the reviewer for raising this useful point. We have clarified our goal regarding phenotypes in the Introduction, and have extensively revised the section "Identifying Perturbations." We believe that the most direct answer is that it is reasonable to consider as "biologically relevant" phenotypes that are statistically significant and when predicted to be observed for as yet untested combinations of drugs and targets match subsequently corrected images.” I strongly disagree with the authors on this point. In my experience, technical artifacts typically show the strongest statistical significance in automated microscopy image analysis. The authors should show convincing examples of the phenotypes they believe are "biologically relevant" or remove the claims that their approach can identify new phenotypes.*

We have clarified how we are using the term “phenotype” at the beginning of the Results, and have added a paragraph to the Discussion to address the reviewers’ concern. Of course, the issue of “biologically relevant” can be problematic. We hope that moving the focus away from a definition of phenotype to a focus on identifying “which drugs have consistent effects on which targets” will be satisfactory. We have added an additional (very large) figure (Figure 5) that displays samples from each phenotype. It illustrates that, at most, a small number of our phenotypes are affected by artifacts. It also allows the reader to assess the extent to which different clusters reflect bona fide pattern differences; note however, as we have rigorously shown many years ago, that it can be very difficult to visually distinguish subcellular patterns that are reproducibly distinguished by numerical features. We note that all of the images collected in our study will be publically available if investigators wish to further examine any specific observed effect.

We are open to advice about this figure, in which very little can be seen unless it is viewed on a computer and magnified. We could make it a supplementary file, or perhaps only show the 31 clusters that are present in more than one drug-clone combination.

*2) I also asked for clarification and improvement of their validation of a new phenotype in Figure 5. The authors have done a good job to clarify the explanation of their independent test of a prediction in Figure 5. However, they still only show one cell in each panel, provide no statistical evidence that these patterns are actually different from the vehicle, nor do they show that the changes observed in these new images are consistent with the images that were used by the active learner. I therefore find this "validation" unconvincing.*

We have done additional independent imaging and completely revised the old Figure 5 (now Figure 6 in the current manuscript). We have extensively modified the text in the Results section that discusses this figure, including showing the statistical significance of the differences.